# MonoDistill: Learning Spatial Features for Monocular 3D Object Detection

**Zhiyu Chong**[*1], **Xinzhu Ma**[*2], **Hong Zhang**[1], **Yuxin Yue**[1], **Haojie Li**[1],
**Zhihui Wang**[1], and **Wanli Ouyang**[2]
[1]Dalian University of Technology    [2]The University of Sydney
`{czydlut, jingshui, 22017036}@mail.dlut.edu.cn`
`{xinzhu.ma, wanli.ouyang}@sydney.edu.au`
`{hjli, zhwang}@dlut.edu.cn`

## Abstract

3D object detection is a fundamental and challenging task for 3D scene understanding, and the monocular-based methods can serve as an economical alternative to the stereo-based or LiDAR-based methods. However, accurately detecting objects in the 3D space from a single image is extremely difficult due to the lack of spatial cues. To mitigate this issue, we propose a simple and effective scheme to introduce the spatial information from LiDAR signals to the monocular 3D detectors, without introducing any extra cost in the inference phase. In particular, we first project the LiDAR signals into the image plane and align them with the RGB images. After that, we use the resulting data to train a 3D detector (LiDAR Net) with the same architecture as the baseline model. Finally, this LiDAR Net can serve as the teacher to transfer the learned knowledge to the baseline model. Experimental results show that the proposed method can significantly boost the performance of the baseline model and ranks the $1^{st}$ place among all monocular-based methods on the KITTI benchmark. Besides, extensive ablation studies are conducted, which further prove the effectiveness of each part of our designs and illustrate what the baseline model has learned from the LiDAR Net. Our code will be released at `https://github.com/monster-ghost/MonoDistill`.

## 1 Introduction

3D object detection is an indispensable component for 3D scene perception, which has wide applications in the real world, such as autonomous driving and robotic navigation. Although the algorithms with stereo (Li et al., 2019b; Wang et al., 2019; Chen et al., 2020a) or LiDAR sensors (Qi et al., 2018; Shi et al., 2019; 2020) show promising performances, the heavy dependence on the expensive equipment restricts the application of these algorithms. Accordingly, the methods based on the cheaper and more easy-to-deploy monocular cameras (Xu & Chen, 2018; Ma et al., 2019; 2021; Brazil & Liu, 2019; Ding et al., 2020) show great potentials and have attracted lots of attention.

As shown in Figure 1 (a), some prior works (Brazil & Liu, 2019; Simonelli et al., 2019; Chen et al., 2020b) directly estimate the 3D bounding boxes from monocular images. However, because of the lack of depth cues, it is extremely hard to accurately detect the objects in the 3D space, and the localization error is the major issue of these methods (Ma et al., 2021). To mitigate this problem, an intuitive idea is to estimate the depth maps from RGB images, and then use them to augment the input data (Figure 1 (b)) (Xu & Chen, 2018; Ding et al., 2020) or directly use them as the input data (Figure 1 (c)) (Wang et al., 2019; Ma et al., 2019). Although these two strategies have made significant improvement in performance, the drawbacks of them can not be ignored: (1) These methods generally use an off-the-shelf depth estimator to generate the depth maps, which introduce lots of computational cost (*e.g.* the most commonly used depth estimator (Fu et al., 2018) need about 400ms to process a standard KITTI image). (2) The depth estimator and detector are trained separately, which may lead to a sub-optimal optimization. Recently, Reading et al. (2021) propose an end-to-end framework (Figure 1 (d)) for monocular 3D detection, which can also leverage depth

---

[*]Equal contribution.

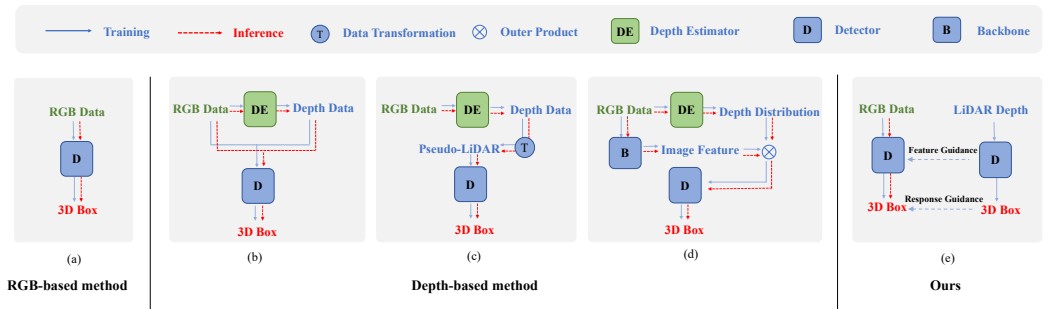

Figure 1: Comparison on the high-level paradigms of monocular 3D detectors.

estimator to provide depth cues. Specifically, they introduce a sub-network to estimate the depth distribution and use it to enrich the RGB features. Although this model can be trained end-to-end and achieves better performance, it still suffer from the low inference speed (630ms per image), mainly caused by the depth estimator and the complicated network architecture. Note that the well-designed monocular detectors, like Ma et al. (2021); Zhang et al. (2021b); Lu et al. (2021), only take about 40ms per image.

In this paper, we aim to introduce the depth cues to the monocular 3D detectors without introducing any extra cost in the inference phase. Inspired by the knowledge distillation (Hinton et al.), which can transfer the learned knowledge from a well-trained CNN to another one without any changes in the model design, we propose that the spatial cues may also be transferred by this way from the LiDAR-based models. However, the main problem for this proposal is the difference of the feature representations used in these two kinds of methods (2D images features *vs.* 3D voxel features). To bridge this gap, we propose to project the LiDAR signals into the image plane and use the 2D CNN, instead of the commonly used 3D CNN or point-wise CNN, to train a 'image-version' LiDAR-based model. After this alignment, the knowledge distillation can be friendly applied to enrich the features of our monocular detector.

Based on the above-mentioned motivation and strategy, we propose the **distill**ation based **mono**cular 3D detector (**MonoDistill**): We first train a teacher net using the projected LiDAR maps (used as the ground truths of the depth estimator in previous works), and then train our monocular 3D detector under the guidance of the teacher net. We argue that, compared with previous works, the proposed method has the following two advantages: First, our method directly learn the spatial cues from the teacher net, instead of the estimated depth maps. This design performs better by avoiding the information loss in the proxy task. Second, our method does not change the network architecture of the baseline model, and thus no extra computational cost is introduced.

Experimental results on the most commonly used KITTI benchmark, where we rank the $1^{st}$ place among all monocular based models by applying the proposed method on a simple baseline, demonstrate the effectiveness of our approach. Besides, we also conduct extensive ablation studies to present each design of our method in detail. More importantly, these experiments clearly illustrate the improvements are achieved by the introduction of spatial cues, instead of other unaccountable factors in CNN.

## 2 RELATED WORKS

**3D detection from only monocular images.** 3D detection from only monocular image data is challenging due to the lack of reliable depth information. To alleviate this problem, lots of scholars propose their solutions in different ways, including but not limited to network design (Roddick et al., 2019; Brazil & Liu, 2019; Zhou et al., 2019; Liu et al., 2020; Luo et al., 2021), loss formulation (Simonelli et al., 2019; Ma et al., 2021), 3D prior (Brazil & Liu, 2019), geometric constraint (Mousavian et al., 2017; Qin et al., 2019; Li et al., 2019a; Chen et al., 2020b), or perspective modeling (Zhang et al., 2021a; Lu et al., 2021; Shi et al., 2021).

**Depth augmented monocular 3D detection.** To provide the depth information to the 3D detectors, several works choose to estimate the depth maps from RGB images. According to the usage of the

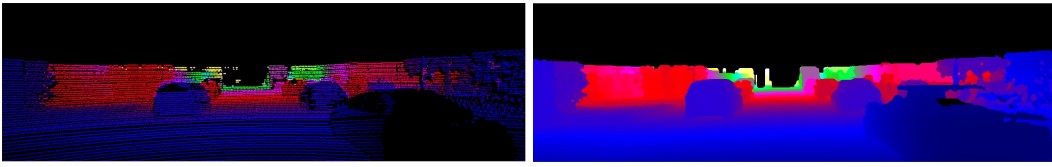

Figure 2: Visualization of the sparse LiDAR maps (*left*) and the dense LiDAR maps (*right*).

estimated depth maps, these methods can be briefly divided into three categories. The first class of these methods use the estimated depth maps to augment the RGB images (Figure 1 (b)). In particular, Xu & Chen (2018) propose three fusion strategies of the RGB images and depth maps, while Ding et al. (2020) and Wang et al. (2021a) focus on how to enrich the RGB features with depth maps in the latent feature space. Besides, Wang et al. (2019); Ma et al. (2019) propose another pipeline (Figure 1 (c)): They back-project the depth maps into the 3D space, and then train a LiDAR-based model and use the resulting data (pseudo-LiDAR signal) to predict the 3D boxes. This framework shows promising performance, and lots of works (Weng & Kitani, 2019; Cai et al., 2020; Wang et al., 2020a; Chu et al., 2021) are built on this solid foundation. Recently, Reading et al. (2021) propose another way to leverage the depth cues for monocular 3D detection (Figure 1 (d)). Particularly, they first estimate the depth distribution using a sub-network, and then use it to lift the 2D features into 3D features, which is used to generate the final results. Compared with the previous two families, this model can be trained in the end-to-end manner, avoiding the sub-optimal optimization. However, a common disadvantage of these methods is that they inevitably increase the computational cost while introducing depth information. Unlike these methods, our model chooses to learn the feature representation under the guidance of depth maps, instead of integrating them. Accordingly, the proposed model not only introduces rich depth cues but also maintains high efficiency.

**Knowledge distillation.** Knowledge distillation (KD) is initially proposed by Hinton et al. for model compression, and the main idea of this mechanism is transferring the learned knowledge from large CNN models to the small one. This strategy has been proved in many computer vision tasks, such as 2D object detection (Dai et al., 2021; Chen et al., 2017; Gupta et al., 2016), semantic semantic segmentation (Hou et al., 2020; Liu et al., 2019). However, few work explore it in monocular 3D detection. In this work, we design a KD-based paradigm to efficiently introduce depth cues for monocular 3D detectors.

**LIGA Stereo.** We found a recent work LIGA Stereo (Guo et al., 2021) (submitted to arXiv on 18 Aug. 2021) discusses the application of KD for stereo 3D detection under the guidance of LiDAR signals. Here we discuss the main differences of LIGA Stereo and our work. First, the tasks and underlying data are different (monocular *vs.* stereo), which leads to different conclusions. For example, Guo et al. (2021) concludes that using the predictions of teacher net as 'soft label' can not bring benefits. However, our experimental results show the effectiveness of this design. Even more, in our task, supervising the student net in the result space is more effective than feature space. Second, they use an off-the-shelf LiDAR-based model to provide guidance to their model. However, we project the LiDAR signals into image plane and use the resulting data to train the teacher net. Except for the input data, the teacher net and student net are completely aligned, including network architecture, hyper-parameters, and training schedule. Third, to ensure the consistent shape of features, LIGA Stereo need to generate the cost volume from stereo images, which is time-consuming (it need about 350ms to estimate 3D boxes from a KITTI image) and hard to achieve for monocular images. In contrast, our method align the feature representations by adjusting the LiDAR-based model, instead of the target model. This design makes our method more efficient (about 35ms per image) and can generalize to all kinds of image-based models in theory.

## 3 METHOD

### 3.1 OVERVIEW

Figure 3 presents the framework of the proposed MonoDistill, which mainly has three components: a monocular 3D detector, an aligned LiDAR-based detector, and several side branches which build

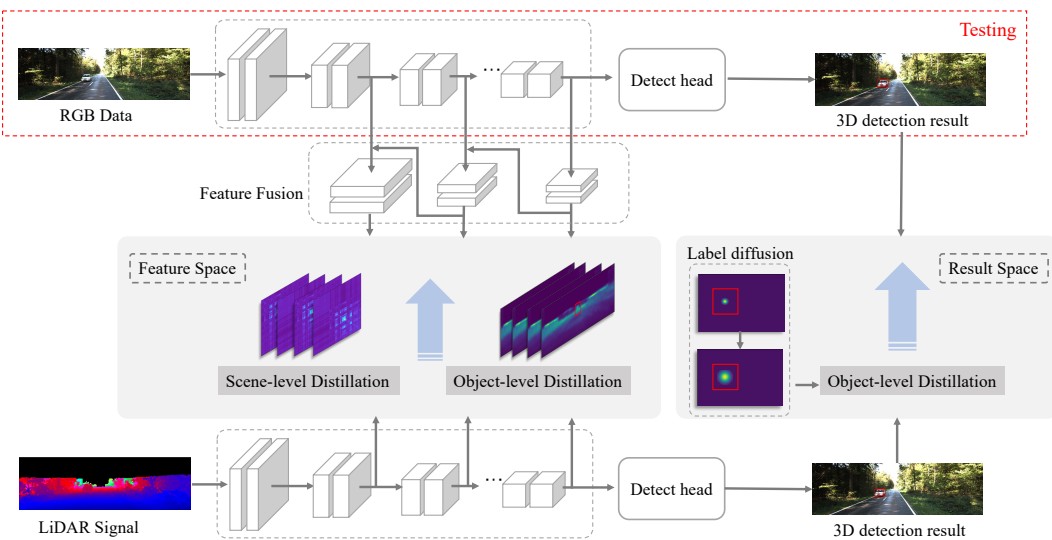

Figure 3: **Illustration of the proposed MonoDistill.** We first generate the 'image-like' LiDAR maps from the LiDAR signals and then train a teacher model using an identical network to the student model. Finally, we propose three distillation schemes to train the student model under the guidance of the well-trained teacher net. In the inference phase, only the student net is used.

the bridge to provide guidance from the LiDAR-based detector to our monocular 3D detector. We will introduce the how to build these parts one by one in the rest of this section.

## 3.2 BASELINE MODEL

**Student model.** We use the one-stage monocular 3D detector MonoDLE (Ma et al., 2021) as our baseline model. Particularly, this baseline model adopts DLA-34 (Yu et al., 2017) as the backbone and uses several parallel heads to predict the required items for 3D object detection. Due to this clean and compact design, this model achieves good performance with high efficiency. Besides, we further normalize the confidence of each predicted object using the estimated depth uncertainty (see Appendix A.1 for more details), which brings about 1 AP improvement *.

**Teacher model.** Existing LiDAR-based models are mainly based on the 3D CNN or point-wise CNN. To align the gap between the feature representations of the monocular detector and the LiDAR-based detector, we project the LiDAR points into the image plane to generate the sparse depth map. Further, we also use the interpolation algorithm (Ku et al., 2018) to generate the dense depth, and see Figure 2 for the visualization of generated data. Then, we use these 'image-like LiDAR maps' to train a LiDAR-based detector using the identical network with our student model.

## 3.3 MONODISTILL

In order to transfer the spatial cues from the well-trained teacher model to the student model, we design three complementary distillation schemes to provide additional guidance to the baseline model.

**Scene-level distillation in the feature space.** First, we think directly enforcing the image-based model learns the feature representations of the LiDAR-based models is sub-optimal, caused by the different modalities. The scene level knowledge can help the monocular 3D detectors build a high-level understanding for the given image by encoding the relative relations of the features, keeping the knowledge structure and alleviating the modality gap. Therefore, we train our student model under the guidance of the high-level semantic features provided by the backbone of the teacher model. To better model the structured cues, we choose to learn the affinity map (Hou et al., 2020) of high-level

---

*Note that both the baseline model and the proposed method adopted the confidence normalization in the following experiments for a fair comparison.

features, instead of the features themselves. Specifically, we first generate the affinity map, which encodes the similarity of each feature vector pair, for both the teacher and student network, and each element $A_{i,j}$ in this affinity map can be computed by:

$$A_{i,j} = \frac{f_i^T f_j}{||f_i||_2 \cdot ||f_j||_2}, \tag{1}$$

where $f_j$ and $f_j$ denote the $i^{th}$ and $j^{th}$ feature vector. After that, we use the L1 norm to enforce the student net to learn the structured information from the teacher net:

$$\mathcal{L}_{sf} = \frac{1}{K \times K} \sum_{i=1}^{K} \sum_{j=1}^{K} ||A_{i,j}^t - A_{i,j}^s||_1, \tag{2}$$

where K is the number of the feature vectors. Note the computational/storage complexity is quadratically related to K. To reduce the cost, we group all features into several local regions and generate the affinity map using the features of local regions. This makes the training of the proposed model more efficient, and we did not observe any performance drop caused by this strategy.

**Object-level distillation in the feature space.** Second, except for the affinity map, directly using the features from teacher net as guidance may also provide valuable cues to the student. However, there is much noise in feature maps, since the background occupies most of the area and is less informative. Distilling knowledge from these regions may make the network deviate from the right optimization direction. To make the knowledge distillation more focused, limiting the distillation area is necessary. Particularly, the regions in the ground-truth 2D bounding boxes are used for knowledge transfer to mitigate the effects of noise. Specifically, given the feature maps of the teacher model and student model $\{F^t, F^s\}$, our second distillation loss can be formulated as.

$$\mathcal{L}_{of} = \frac{1}{N_{pos}} ||M_{of}(F_s - F_t)||_2^2, \tag{3}$$

where $M_{of}$ is the mask generated from the center point and the size of 2D bounding box and $N_{pos}$ is the number of valid feature vectors.

**Object-level distillation in the result space.** Third, similar to the traditional KD, we use the predictions from the teacher net as extra 'soft label' for the student net. Note that in this scheme, only the predictions on the foreground region should be used, because the predictions on the background region are usually false detection. As for the definition of the 'foreground regions', inspired by CenterNet (Zhou et al., 2019), a simple baseline is regarding the center point as the foreground region. Furtherly, we find that the quality of the predicted value of the teacher net near the center point is good enough to guide the student net. Therefore, we generate a Gaussian-like mask (Tian et al., 2019; Wang et al., 2021b) based on the position of the center point and the size of 2D bounding box and the pixels whose response values surpass a predefined threshold are sampled, and then we train these samples with equal weights (see Figure 4 for the visualization). After that, our third distillation loss can be formulated as:

$$\mathcal{L}_{or} = \sum_{k=1}^{N} ||M_{or}(y_k^s - y_k^t)||_1, \tag{4}$$

where $M_{or}$ is the mask which represents positive and negative samples, $y_k$ is the output of the $k^{th}$ detection head and $N$ is the number of detection heads.

**Additional strategies.** We further propose some strategies for our method. First, for the distillation schemes in the feature space (*i.e.* $\mathcal{L}_{sf}$ and $\mathcal{L}_{of}$), we only perform them on the last three blocks of the backbone. The main motivation of this strategy is: The first block usually is rich in the low-level features (such as edges, textures, etc.). The expression forms of the low-level features for LiDAR and image data may be completely different, and enforcing the student net to learn these features in a modality-across manner may mislead it. Second, in order to better guide the student to learn spatial-aware feature representations, we apply the attention based fusion module (FF in Table 1) proposed by Chen et al. (2021b) in our distillation schemes in the feature space ( i.e. $\mathcal{L}_{sf}$ and $\mathcal{L}_{of}$).

**Loss function.** We train our model in an end-to-end manner using the following loss function:

$$\mathcal{L} = \mathcal{L}_{src} + \lambda_1 \cdot \mathcal{L}_{sf} + \lambda_2 \cdot \mathcal{L}_{of} + \lambda_3 \cdot \mathcal{L}_{or}, \tag{5}$$

where $\mathcal{L}_{src}$ denotes the loss function used in the MonoDLE (Ma et al., 2021). $\lambda_1, \lambda_2, \lambda_3$ are the hyper-parameters to balance each loss. For the teacher net, only $\mathcal{L}_{src}$ is adopted.

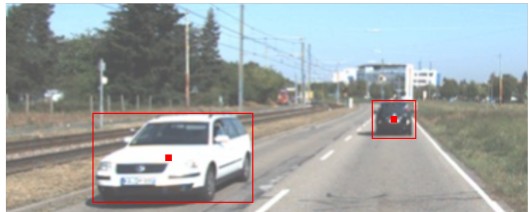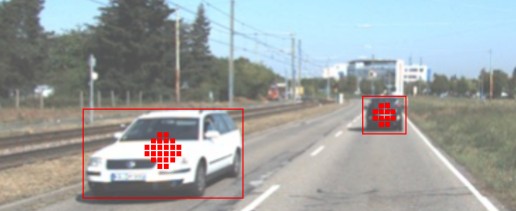

Figure 4: *Left:* Regard the center point as the foreground region. *Right:* Generate foreground region from the center point and the size of bounding box. Besides, the 2D bounding boxes are used as the foreground region for $\mathcal{L}_{\mathbf{of}}$.

## 4 EXPERIMENTS

### 4.1 SETUP

**Dataset and metrics.** We conduct our experiments on the KITTI (Geiger et al., 2012), which is most commonly used dataset in 3D detection task. Specifically, this dataset provides 7,481 training samples and 7,518 testing samples, and we further divide the training data into a train set (3,712 samples) and a validation set (3,769 samples), following prior works (Chen et al., 2015). Both 3D detection and Bird's Eye View (BEV) detection are evaluated using $\text{AP}|_{\text{R}_{40}}$ (Simonelli et al., 2019) as metric. We report our final results on the *testing* set, while the ablation studies are conducted on the *validation* set. Besides, we mainly focus on the **Car** category, while also present the performances of **Pedestrian** and **Cyclist** in Appendix A.2 for reference.

**Implementation.** We provide the implementation details in Appendix A.1. Besides, our code will be open-sourced for the reproducibility.

### 4.2 MAIN RESULTS

**Ablation studies.** Table 1 shows the ablation studies of the proposed methods. Specifically, we found that all three distillation schemes can improve the accuracy of the baseline model, and the improvements of them are complementary. Besides, the feature fusion strategy can also boost the accuracy. Compared with the baseline, our full model improves 3D detection performance by **3.34**, **5.02**, **2.98** and improve BEV performance by **5.16**, **6.62**, **3.87** on the moderate, easy and hard settings respectively.

Table 1: **Ablation studies on the KITTI *validation* set.** SF, OF, and OR denote the scene-level distillation in feature space, the object-level distillation in feature space, and the object-level distillation in result space, respectively. Besides, FF means the attention based feature fusion strategy.

| | SF | OF | OR | FF | 3D@IOU=0.7 | | | BEV@IOU=0.7 | | |
|---|---|---|---|---|---|---|---|---|---|---|
| | | | | | Mod. | Easy | Hard | Mod. | Easy | Hard |
| a. | | | | | 15.13 | 19.29 | 12.78 | 20.24 | 26.47 | 18.29 |
| b. | ✓ | | | | 16.96 | 21.99 | 14.42 | 22.79 | 29.76 | 19.78 |
| c. | | ✓ | | | 16.85 | 21.76 | 14.36 | 22.30 | 28.93 | 19.31 |
| d. | | | ✓ | | 17.24 | 21.63 | 14.71 | 23.47 | 30.52 | 20.33 |
| e. | ✓ | ✓ | | | 17.33 | 22.34 | 14.63 | 22.90 | 30.02 | 19.84 |
| f. | ✓ | | ✓ | | 17.70 | 22.59 | 15.17 | 23.59 | 31.07 | 20.46 |
| g. | | ✓ | ✓ | | 17.98 | 22.58 | 15.26 | 23.76 | 30.98 | 20.52 |
| h. | ✓ | ✓ | ✓ | | 18.24 | 23.82 | 15.49 | 25.06 | 32.66 | 21.88 |
| i. | ✓ | ✓ | ✓ | ✓ | **18.47** | **24.31** | **15.76** | **25.40** | **33.09** | **22.16** |

**Detailed design choice.** We provide additional experiments in Table 2 for our method. First, as for object-level distillation in the feature space, we investigate the different effects of applying distillation on the whole image and foreground regions. Due to the noise in the background, guiding the foreground regions is more effective than the whole image, which improves the accuracy by 0.72 on

the moderate settings in 3D detection. Second, as for object-level distillation in the result space, we compare the different effects of point label and region label. It can be observed that the generated region can significantly increase performance while guiding only in sparse point label brings limited improvements. Our proposed label diffusion strategy can increase the number of positive samples for supervision, thus improving performance.

Table 2: **Evaluation on the KITTI *validation* set for detailed design choice.** OF and OR represent the object-level distillation in feature space and the object-level distillation in result space.

| Guidance | Choice | 3D@IOU=0.7 | | | BEV@IOU=0.7 | | |
|---|---|---|---|---|---|---|---|
| | | Mod. | Easy | Hard | Mod. | Easy | Hard |
| OF | full | 16.13 | 21.52 | 14.18 | 22.04 | 27.85 | 19.04 |
| | foreground | 16.85 | 21.76 | 14.36 | 22.30 | 28.93 | 19.31 |
| OR | sparse label | 15.51 | 20.58 | 13.70 | 21.47 | 27.16 | 18.60 |
| | diffused label | 17.24 | 21.63 | 14.71 | 23.47 | 30.52 | 20.33 |

**Comparison with state-of-the-art methods.** Table 3 and Table 4 compare the proposed method with other state-of-the-art methods on the KITTI *test* and *validation* sets. On the *test* set, the proposed method outperforms existing methods in all metrics. We note that, compared with previous best results, we can obtain **1.83, 0.50, 1.53** improvements on the moderate, easy and hard settings in 3D detection. Furthermore, our method achieves more significant improvements in BEV detection, increasing upon the prior work by **2.51, 1.21, 2.46** on the moderate, easy and hard settings. Moreover, compared with the depth-based methods, our method outperforms them in performance by a margin and is superior to theirs in the inference speed. By contrast, our method only takes 40ms to process a KITTI image, tested on a single NVIDIA GTX 1080Ti, while the Fastest of the depth-based methods (Ma et al., 2019; 2020; Ding et al., 2020; Wang et al., 2021a; Reading et al., 2021) need 180ms. On the *validation* set, the proposed also performs best, both for the 0.7 IoU threshold and 0.5 IoU threshold. Besides, we also present the performance of the baseline model to better show the effectiveness of the proposed method. Note that we do not report the performances of some depth-based methods (Ma et al., 2019; 2020; Ding et al., 2020; Wang et al., 2021a) due to the data leakage problem [*].

### 4.3 More Discussions

**What has the student model learned from the teacher model?** To locate the source of improvement, we use the items predicted from the baseline model to replace that from our full model, and Table 5 summarizes the results of the cross-model evaluation. From these results, we can see that the teacher model provides effective guidance to the location estimation (b→f), and improvement of dimension part is also considerable (c→f). Relatively, the teacher model provides limited valuable cues to the classification and orientation part. This phenomenon suggests the proposed methods boost the performance of the baseline model mainly by introducing the spatial-related information, which is consistent with our initial motivation. Besides, we also show the errors of depth estimation, see Appendix A.3 for the results.

**Is the effectiveness of our method related to the performance of the teacher model?** An intuitive conjecture is the student can learn more if the teacher network has better performance. To explore this problem, we also use the sparse LiDAR maps to train a teacher net to provide guidance to the student model (see Figure 2 for the comparison of the sparse and dense data). As shown in Table 6, the performance of the teacher model trained from the sparse LiDAR maps is largely behind by that from dense LiDAR maps (drop to 22.05% from 42.45%, moderate setting), while both of them provides comparable benefits to the student model. Therefore, for our task, the performance of the teacher model is not directly related to the performance improvement, while the more critical factor is whether the teacher network contains complementary information to the student network.

**Do we need depth estimation as an intermediate task?** As shown in Figure 1, most previous methods choose to estimate the depth maps to provide depth information for monocular 3D detection (information flow: LiDAR data →estimated depth map→3D detector). Compared with this

---

[*] these methods use the depth estimator pre-trained on the KITTI Depth, which overlaps with the validation set of the KITTI 3D.

Table 3: **Comparison of state-of-the-art methods on the KITTI *test* set.** Methods are ranked by moderate setting. We highlight the best results in **bold** and the second place in underlined. Only RGB images are required as input in the inference phase for all listed methods. *: need dense depth maps or LiDAR signals for training. †: our baseline model without confidence normalization.

| Method | 3D@IOU=0.7 | | | BEV@IOU=0.7 | | | Runtime |
|---|---|---|---|---|---|---|---|
| | Mod. | Easy | Hard | Mod. | Easy | Hard | |
| M3D-RPN (Brazil & Liu, 2019) | 9.71 | 14.76 | 7.42 | 13.67 | 21.02 | 10.23 | 160 ms |
| SMOKE (Liu et al., 2020) | 9.76 | 14.03 | 7.84 | 14.49 | 20.83 | 12.75 | 30 ms |
| MonoPair (Chen et al., 2020b) | 9.99 | 13.04 | 8.65 | 14.83 | 19.28 | 12.89 | 60 ms |
| RTM3D (Li et al., 2020) | 10.34 | 14.41 | 8.77 | 14.20 | 19.17 | 11.99 | 50 ms |
| AM3D* (Ma et al., 2019) | 10.74 | 16.50 | 9.52 | 17.32 | 25.03 | 14.91 | 400 ms |
| PatchNet* (Ma et al., 2020) | 11.12 | 15.68 | 10.17 | 16.86 | 22.97 | 14.97 | 400 ms |
| D4LCN* (Ding et al., 2020) | 11.72 | 16.65 | 9.51 | 16.02 | 22.51 | 12.55 | 200 ms |
| MonoDLE† (Ma et al., 2021) | 12.26 | 17.23 | 10.29 | 18.89 | 24.79 | 16.00 | 40 ms |
| MonoRUn* (Chen et al., 2021a) | 12.30 | 19.65 | 10.58 | 17.34 | 27.94 | 15.24 | 70 ms |
| GrooMeD-NMS (Kumar et al., 2021) | 12.32 | 18.10 | 9.65 | 18.27 | 16.19 | 14.05 | 120 ms |
| DDMP-3D* (Wang et al., 2021a) | 12.78 | 19.71 | 9.80 | 17.89 | 28.08 | 13.44 | 180 ms |
| CaDDN* (Reading et al., 2021) | 13.41 | 19.17 | 11.46 | 18.91 | 27.94 | 17.19 | 630 ms |
| MonoEF (Zhou et al., 2021) | 13.87 | 21.29 | 11.71 | 19.70 | 29.03 | 17.26 | 30 ms |
| MonoFlex (Zhang et al., 2021b) | 13.89 | 19.94 | 12.07 | 19.75 | 28.23 | 16.89 | 30 ms |
| Autoshape (Liu et al., 2021) | 14.17 | 22.47 | 11.36 | 20.08 | 30.66 | 15.59 | 50 ms |
| GUPNet (Lu et al., 2021) | 14.20 | 20.11 | 11.77 | - | - | - | 35 ms |
| Ours* | **16.03** | **22.97** | **13.60** | **22.59** | **31.87** | **19.72** | 40 ms |
| Improvements | +1.83 | +0.50 | +1.53 | +2.51 | +1.21 | +2.46 | - |

Table 4: **Performance of the Car category on the KITTI *validation* set.** We highlight the best results in **bold** and the second place in underlined. †: our baseline model without confidence normalization.

| Method | 3D@IOU=0.7 | | | BEV@IOU=0.7 | | | 3D@IOU=0.5 | | | BEV@IOU=0.5 | | |
|---|---|---|---|---|---|---|---|---|---|---|---|---|
| | Mod. | Easy | Hard | Mod. | Easy | Hard | Mod. | Easy | Hard | Mod. | Easy | Hard |
| M3D-RPN | 11.07 | 14.53 | 8.65 | 15.62 | 20.85 | 11.88 | 35.94 | 48.53 | 28.59 | 39.60 | 53.35 | 31.76 |
| MonoPair | 12.30 | 16.28 | 10.42 | 18.17 | 24.12 | 15.76 | 42.39 | 55.38 | 37.99 | 47.63 | 61.06 | 41.92 |
| MonoDLE† | 13.66 | 17.45 | 11.68 | 19.33 | 24.97 | 17.01 | 43.42 | 55.41 | 37.81 | 46.87 | 60.73 | 41.89 |
| GrooMeD-NMS | 14.32 | 19.67 | 11.27 | 19.75 | 27.38 | 15.92 | 43.42 | 55.62 | 32.89 | 44.98 | 61.83 | 36.29 |
| MonoRUn | 14.65 | 20.02 | 12.61 | - | - | - | 43.39 | 59.71 | 38.44 | - | - | - |
| GUPNet | 16.46 | 22.76 | 13.72 | 22.94 | 31.07 | 19.75 | 42.33 | 57.62 | 37.59 | 47.06 | 61.78 | 40.88 |
| MonoFlex | 17.51 | 23.64 | 14.83 | - | - | - | - | - | - | - | - | - |
| Baseline | 15.13 | 19.29 | 12.78 | 20.24 | 26.47 | 18.29 | 43.54 | 57.43 | 39.22 | 48.49 | 63.56 | 42.81 |
| Ours | **18.47** | **24.31** | **15.76** | **25.40** | **33.09** | **22.16** | **49.35** | **65.69** | **43.49** | **53.11** | **71.45** | **46.94** |

Table 5: **Cross-model evaluation on the KITTI *validation* set.** We extract each required item (location, dimension, orientation, and confidence) from the baseline model (B) and the full model (O), and evaluate them in a cross-model manner.

| | loc. | dim. | ori. | con. | 3D@IOU=0.7 | | | BEV@IOU=0.7 | | |
|---|---|---|---|---|---|---|---|---|---|---|
| | | | | | Mod. | Easy | Hard | Mod. | Easy | Hard |
| a. | B | B | B | B | 15.13 | 19.29 | 12.78 | 20.24 | 26.47 | 18.29 |
| b. | B | O | O | O | 16.05 | 20.07 | 13.47 | 21.31 | 27.77 | 19.14 |
| c. | O | B | O | O | 17.91 | 22.87 | 15.29 | 25.09 | 32.78 | 21.93 |
| d. | O | O | B | O | 18.12 | 24.02 | 15.34 | 25.02 | 32.85 | 21.84 |
| e. | O | O | O | B | 18.41 | 24.27 | 15.55 | 24.98 | 32.78 | 21.81 |
| f. | O | O | O | O | 18.47 | 24.31 | 15.76 | 25.40 | 33.09 | 22.16 |

scheme, our method directly learns the depth cues from LiDAR-based methods (information flow: LiDAR data→3D detector), avoiding the information loss in the depth estimation step. Here we quantitatively show the information loss in depth estimation using a simple experiment. Specifically, we use DORN (Fu et al., 2018) (same as most previous depth augmented methods) to generate the depth maps, and then use them to train the teacher net. Table 7 shows the results of this experi-

Table 6: **Performance of the student model under the guidance of different teacher models.** Metric is the $\text{AP}|_{40}$ for the 3D detection task on the KITTI *validation* set. We also show the performance improvements of the student model to the baseline model for better comparison.

|  | Teacher Model | | | Student Model | | | Improvement | | |
|---|---|---|---|---|---|---|---|---|---|
|  | Mod. | Easy | Hard | Mod. | Easy | Hard | Mod. | Easy | Hard |
| sparse maps | 22.05 | 31.67 | 18.72 | 18.07 | 23.61 | 15.36 | +2.94 | +4.32 | +2.58 |
| dense maps | 42.57 | 58.06 | 37.07 | 18.47 | 24.31 | 15.76 | +3.34 | +5.02 | +2.98 |

ment. Note that, compared with setting c, setting b's teacher net is trained from a larger training set (23,488 *vs.* 3,712) with ground-truth depth maps (ground truth depth maps vs. noisy depth maps). Nevertheless, this scheme still lags behind our original method, which means that there is serious information loss in monocular depth estimation (stereo image performs better, which is discussed in Appendix A.4).

Table 7: **Comparison of using depth estimation as intermediate task or not**. Setting a. and c. denote the baseline model and our full model. Setting b. uses the depth maps generated from DORN (Fu et al., 2018) to train the teacher model. Experiments are conducted on the KITTI *validation* set.

|  | 3D@IOU=0.7 | | | BEV@IOU=0.7 | | | AOS@IOU=0.7 | | | 2D@IOU=0.7 | | |
|---|---|---|---|---|---|---|---|---|---|---|---|---|
|  | Mod. | Easy | Hard | Mod. | Easy | Hard | Mod. | Easy | Hard | Mod. | Easy | Hard |
| a. | 15.13 | 19.29 | 12.78 | 20.24 | 26.47 | 18.29 | 90.95 | 97.46 | 83.02 | 92.18 | 98.37 | 85.05 |
| b. | 17.70 | 23.21 | 15.02 | 23.34 | 31.20 | 20.40 | 91.50 | 97.77 | 83.49 | 92.51 | 98.54 | 85.38 |
| c. | 18.47 | 24.31 | 15.76 | 25.40 | 33.09 | 22.16 | 91.67 | 97.88 | 83.59 | 92.71 | 98.58 | 85.56 |

## 4.4 QUALITATIVE RESULTS

In Figure 5, we show the qualitative comparison of detection results. We can see that the proposed method shows better localization accuracy than the baseline model. See Appendix A.7 for more detailed qualitative results.

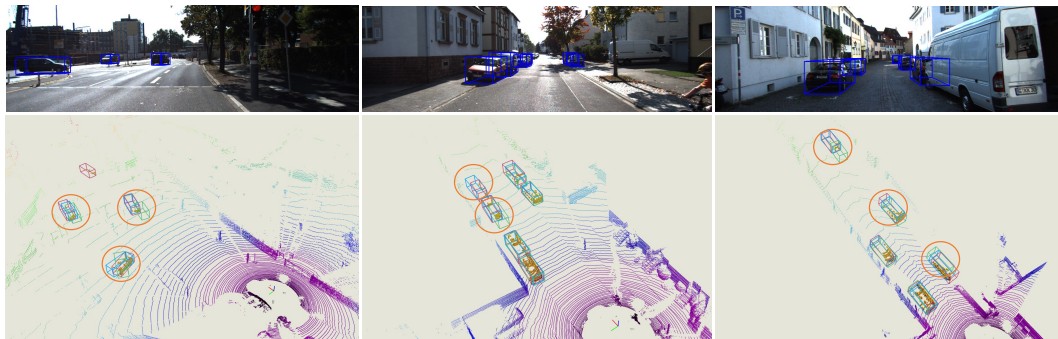

Figure 5: **Qualitative results**. We use green, blue and red boxes to denote the results from baseline, our method, and ground truth. Besides, we use red circle to highlight the main differences.

## 5 CONCLUSION

In this work, we propose the MonoDistill, which introduces spatial cues to the monocular 3D detector based on the knowledge distillation mechanism. Compared with previous schemes, which share the same motivation, our method avoids any modifications on the target model and directly learns the spatial features from the model rich in these features. This design makes the proposed method perform well in both performance and efficiency. To show an all-around display of our model, extensive experiments are conducted on the KITTI dataset, where the proposed method ranks $1^{st}$ at 25 FPS among all monocular 3D detectors.

## ACKNOWLEDGEMENTS

This work was supported in part by the National Natual Science Foundation of China (NSFC) under Grants No.61932020, 61976038, U1908210 and 61772108. Wanli Ouyang was supported by the Australian Research Council Grant DP200103223, FT210100228, and Australian Medical Research Future Fund MRFAI000085.

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

# A    APPENDIX

## A.1    MORE DETAILS OF THE BASELINE MODEL

**Network architecture.** The baseline network is extended from the anchor-free 2D object detection framework, which consists of a feature extraction network and seven detection subheads. We employ DLA-34 (Yu et al., 2017) without deformable convolutions as our backbone. The feature maps are downsampled by 4 times and then we take the image features as input and use 3x3 convolution, ReLU, and 1x1 convolution to output predictions for each detection head. Detection head branches include three for 2D components and four for 3D components. Specifically, 2D detection heads include heatmap, offset between the 2D key-point and the 2D box center, and size of 2D box. 3D components include offset between the 2D key-point and the projected 3D object center, depth, dimensions, and orientations. As for objective functions, we train the heatmap with focal loss. The other loss items adopt L1 losses except for depth and orientation. The depth branch employs a modified L1 loss with the assist of heteroscedastic aleatoric uncertainty. Common *MultiBin* loss is used for the orientation branch. Besides, we propose a strategy to improve the accuracy of baseline. Inspire by (Lu et al., 2021), estimated depth uncertainty can provide confidence for each projection depth. Therefore, we normalize the confidence of each predicted box using depth uncertainty. In this way, the score has capability of indicating the uncertainty of depth.

**Training details.** Our model is trained on 2 NVIDIA 1080Ti GPUs in an end-to-end manner for 150 epochs. We employ the common Adam optimizer with initial learning rate $1.25e^{-4}$, and decay it by ten times at 90 and 120 epochs. To stabilize the training process, we also applied the warm-up strategy (5 epochs). As for data augmentations, only random random flip and center crop are applied. Same as the common knowledge distillation scheme, we first train teacher network in advance, and then fix the teacher network. As for student network, we simply train the detection model to give a suitable initialization. We implemented our method using PyTorch. And our code is based on Ma et al. (2021).

## A.2    PEDESTRIAN/CYCLIST DETECTION.

Due to the small sizes, non-rigid structures, and limited training samples, the pedestrians and cyclists are much more challenging to detect than cars. We first report the detection results on *test* set in Table 8. It can be seen that our proposed method is also competitive with current state-of-the-art methods on the KITTI *test* set, which increases 0.69 AP on hard difficulty level of pedestrian category. Note that, the accuracy of these difficult categories fluctuates greatly compared with Car detection due to insufficient training samples (see Table 10 for the details). Due the access to the test server is limited, we conduct more experiments for pedestrian/cyclist on the *validation* set for general conclusions (we run the proposed method three times with different random seeds), and the experimental results are summarized in Table 9. According to these results, we can find that the proposed method can effectively boost the accuracy of the baseline model for pedestrian/cyclist detection.

Table 8: **Performance of Pedestrian/Cyclist detection on the KITTI *test* set.** We highlight the best results in **bold** and the second place in underlined.

| Method | Pedestrian | | | Cyclist | | |
|---|---|---|---|---|---|---|
| | Easy | Mod. | Hard | Easy | Mod. | Hard |
| M3D-RPN | 4.92 | 3.48 | 2.94 | 0.94 | 0.65 | 0.47 |
| D4LCN | 4.55 | 3.42 | 2.83 | 2.45 | 1.67 | 1.36 |
| MonoPair | 10.02 | 6.68 | 5.53 | 3.79 | 2.21 | 1.83 |
| MonoFlex | 9.43 | 6.31 | 5.26 | 4.17 | 2.35 | 2.04 |
| MonoDLE | 9.64 | 6.55 | 5.44 | 4.59 | 2.66 | 2.45 |
| CaDDN | **12.87** | 8.14 | 6.76 | **7.00** | **3.14** | **3.30** |
| DDMP-3D | 4.93 | 3.55 | 3.01 | 4.18 | 2.50 | 2.32 |
| AutoShape | 5.46 | 3.74 | 3.03 | 5.99 | 3.06 | 2.70 |
| Ours | 12.79 | **8.17** | **7.45** | 5.53 | 2.81 | 2.40 |

Table 9: **Performance of Pedestrian/Cyclist detection on the KITTI *validation* set.** Both 0.25 and 0.5 IoU thresholds are considered. We report the mean of several experiments for the proposed methods. ± captures the standard deviation over random seeds.

| | Method | 3D@IoU=0.25 | | | 3D@IoU=0.5 | | |
|---|---|---|---|---|---|---|---|
| | | Easy | Mod. | Hard | Easy | Mod. | Hard |
| Pedestrian | Baseline | 29.07±0.21 | 23.77±0.15 | 19.85±0.14 | 6.8±0.28 | 5.17±0.08 | 4.37±0.15 |
| | Ours | 32.09±0.71 | 25.53±0.55 | 21.15±0.79 | 8.95±1.26 | 6.84±0.81 | 5.32±0.75 |
| Cyclist | Baseline | 21.06±0.46 | 11.87±0.19 | 10.77±0.02 | 3.71±0.49 | 1.88±0.23 | 1.64±0.04 |
| | Ours | 24.26±1.29 | 13.04±0.44 | 12.08±0.68 | 5.38±0.91 | 2.67±0.40 | 2.53±0.38 |

Table 10: **Training samples** of each category on the KITTI *training* set.

| | cars | pedestrians | cyclists |
|---|---|---|---|
| # instances | 14,357 | 2,207 | 734 |

## A.3 DEPTH ERROR ANALYSIS

As shown in Figure 6, we compare the depth error between baseline and our method. Specifically, we project all valid samples of the Car category into the image plane to get the corresponding predicted depth values. Then we fit the depth errors between ground truths and predictions as a linear function by least square method. According to the experimental results, we can find that our proposed method can boost the accuracy of depth estimation at different distances.

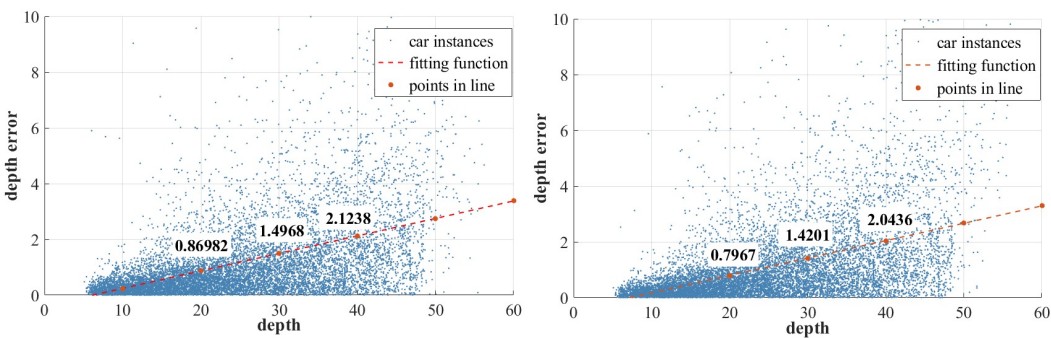

Figure 6: **Errors of depth estimation.** We show the errors of depth estimation as a function of the depth (x-axis) for the baseline model (*left*) and our full model (*right*).

## A.4 THE EFFECTS OF STEREO DEPTH

We also explored the changes in performance under the guidance of estimated stereo depth (Chang & Chen, 2018), and show the results in Table 11. Stereo depth estimation exploits geometric constraints in stereo images to obtain the absolute depth value through pixel-wise matching, which is more accurate compared with monocular depth estimation. Therefore, under the guidance of stereo depth, the model achieves almost the same accuracy as LiDAR signals guidance at 0.5 IoU threshold, and there is only a small performance drop at 0.7 IoU threshold.

## A.5 GENERALIZATION OF THE PROPOSED METHOD

In the main paper, we introduced the proposed method based on MonoDLE (Ma et al., 2021). Here we discuss the generalization ability of the proposed method.

**Generalizing to other baseline models.** To show the generalization ability of the proposed method, we apply our method on another monocular detector GUPNet (Lu et al., 2021), which is a two-stage detection method. Experimental results are shown in the Table 12. We can find that the proposed

Table 11: **Effects of stereo depth estimation.** Baseline denotes the baseline model without guidance of teacher network. Stereo Depth and LiDAR Depth denote under the guidance of stereo depth maps and LiDAR signals. Experiments are conducted on the KITTI *validation* set.

| | 3D@IOU=0.7 | | | BEV@IOU=0.7 | | | 3D@IOU=0.5 | | | BEV@IOU=0.5 | | |
|---|---|---|---|---|---|---|---|---|---|---|---|---|
| | Mod. | Easy | Hard | Mod. | Easy | Hard | Mod. | Easy | Hard | Mod. | Easy | Hard |
| Baseline | 15.13 | 19.29 | 12.78 | 20.24 | 26.47 | 18.29 | 43.54 | 57.43 | 39.22 | 48.49 | 63.56 | 42.81 |
| Stereo Depth | 18.18 | 23.54 | 15.42 | 24.89 | 32.26 | 21.64 | 49.13 | 65.18 | 43.29 | 52.88 | 69.47 | 46.72 |
| LiDAR Depth | 18.47 | 24.31 | 15.76 | 25.40 | 33.09 | 22.16 | 49.35 | 65.69 | 43.49 | 53.11 | 71.45 | 46.94 |

method can also boosts the performances of GUPNet, which confirms the generalization of our method.

Table 12: **MonoDistill on GUPNet.** Experiments are conducted on the KITTI *validation* set.

| | 3D@IOU=0.7 | | | 3D@IOU=0.5 | | |
|---|---|---|---|---|---|---|
| | Easy | Mod. | Hard | Easy | Mod. | Hard |
| GUPNet-Baseline | 22.76 | 16.46 | 13.72 | 57.62 | 42.33 | 37.59 |
| GUPNet-Ours | 24.43 | 16.69 | 14.66 | 61.72 | 44.49 | 40.07 |

**Generalizing to sparse LiDAR signals.** We also explore the changes in performance under different resolution of LiDAR signals. In particular, following Pseudo-LiDAR++ (You et al., 2020), we generate the simulated 32-beam/16-beam LiDAR signals and use them to train our teacher model (in the 'sparse' setting). We show the experimental results, based on MonoDLE, in the Table 13. We can see that, although the improvement is slightly reduced due to the decrease of the resolution of LiDAR signals, the proposed method significantly boost the performances of baseline model under all setting.

Table 13: **Effects of the resolution of LiDAR signals.** Experiments are conducted on the KITTI *validation* set.

| | 3D@IOU=0.7 | | | 3D@IOU=0.5 | | |
|---|---|---|---|---|---|---|
| | Mod. | Easy | Hard | Mod. | Easy | Hard |
| Baseline | 19.29 | 15.13 | 12.78 | 43.54 | 57.43 | 39.22 |
| Ours - 16-beam | 22.49 | 17.66 | 15.08 | 49.39 | 65.45 | 43.60 |
| Ours - 32-beam | 23.24 | 17.71 | 15.19 | 49.41 | 65.61 | 43.46 |
| Ours - 64-beam | 23.61 | 18.07 | 15.36 | 49.67 | 65.97 | 43.74 |

**More discussion.** Besides, note that the camera parameters of the images on the KITTI *test* set are different from these of the *training/validation* set, and the good performance on the *test* set suggests the proposed method can also generalize to different camera parameters. However, generalizing to the new scenes with different statistical characteristics is a hard task for existing 3D detectors (Yang et al.; Wang et al., 2020b), including the image-based models and LiDAR-based models, and deserves further investigation by future works. We also argue that the proposed method can generalize to the new scenes better than other monocular models because ours model learns the stronger features from the teacher net. These results and analysis will be included in the revised version.

A.6   COMPARISON WITH DIRECT DENSE DEPTH SUPERVISION.

According to the ablation studies in the main paper, we can find that depth cues are the key factor to affect the performance of the monocular 3D models. However, dense depth supervision in the student model without KD may also introduce depth cues to the monocular 3D detectors. Here we conduct the control experiment by adding a new depth estimation branch, which is supervised by the dense LiDAR maps. Note that, this model is trained without KD. Table 14 compares the performances of the baseline model, the new control experiment, and the proposed method. From these results, we can get the following conclusions: (i) additional depth supervision can introduce

the spatial cues to the models, thereby improving the overall performance; (ii) the proposed KD-based method significantly performs better than the baseline model and the new control experiment, which demonstrates the effectiveness of our method.

Table 14: **Comparison with direct dense depth supervision.** Experiments are conducted on the KITTI *validation* set.

|  | 3D@IOU=0.7 | | | 3D@IOU=0.5 | | |
|---|---|---|---|---|---|---|
|  | Mod. | Easy | Hard | Mod. | Easy | Hard |
| Baseline | 15.13 | 19.29 | 12.78 | 43.54 | 57.43 | 39.22 |
| Baseline + depth supv. | 17.05 | 21.85 | 14.54 | 46.19 | 60.42 | 41.88 |
| Ours | 18.47 | 24.31 | 15.76 | 49.35 | 65.69 | 43.49 |

## A.7 MORE QUALITATIVE RESULTS

In Figure 7, we show more qualitative results on the KITTI dataset. We use orange box, green, and purple boxes for cars, pedestrians, and cyclists, respectively. In Figure 8, we show comparison of detection results in the 3D space. It can be found that our method can significantly improve the accuracy of depth estimation compared with the baseline.

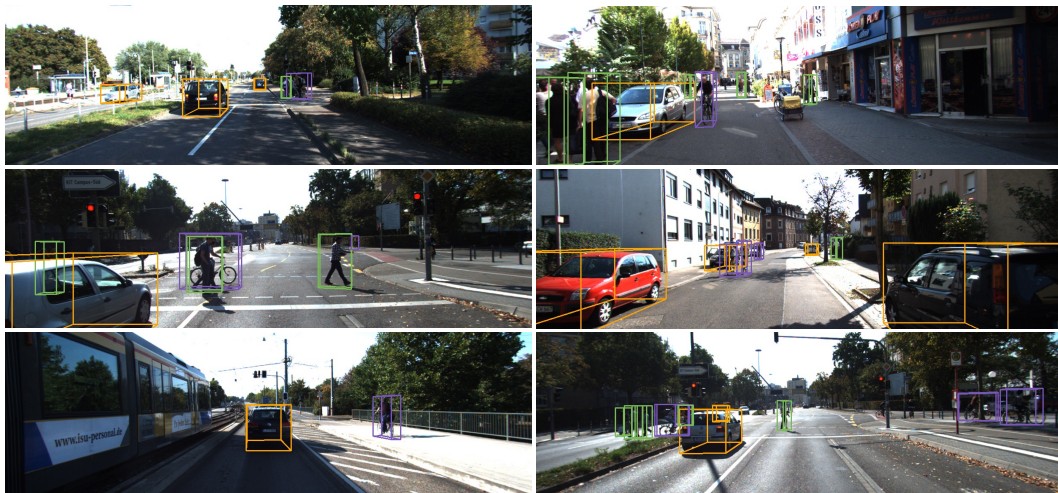

Figure 7: Qualitative results for multi-class 3D object detection. The boxes' color of cars, pedestrian, and cyclist are in orange, green, and purple, respectively.

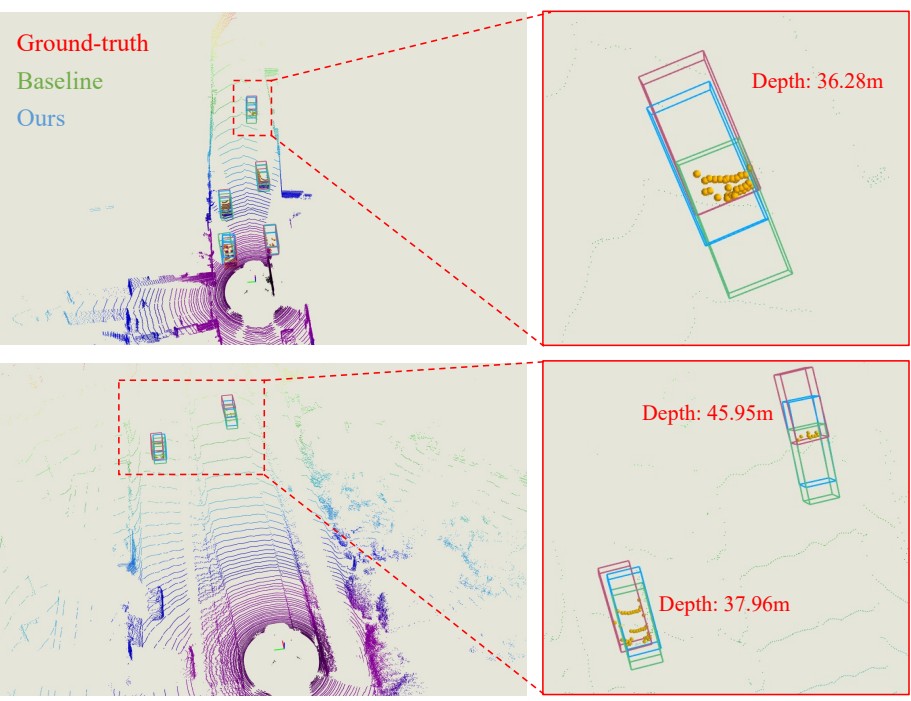

Figure 8: Qualitative results of our method for 3D space. The boxes' color of ground truth, baseline, and ours are in red, green, and blue, respectively.

