# OpenReview forum: "MonoDistill: Learning Spatial Features for Monocular 3D Object Detection"
_ICLR.cc/2022/Conference — ICLR 2022 Poster_

### Official Review · Reviewer_J4VF · 2021-10-18

**Correctness:** 3
**Technical Novelty And Significance:** 3
**Empirical Novelty And Significance:** 3
**Recommendation:** 5
**Confidence:** 5

**Main Review:**

1. This paper claims to be a mono-based 3D object detection method. Although only RGB image data is needed in the test process, point cloud data is actually used in the training process. From this point of view, this claim is not appropriate.

2. Figure 3 is very confusing, please redraw it according to the actual process in order to show the actual process more clearly.

3. This paper claims in many places that the method based on monocular image is more valuable in application than the method based on pure point cloud, but it has to be admitted in the detection result that the result of 3D object detection based on image is far inferior to that based on pure point. So from this point of view, do the above remarks lack the measurement dimension of detection accuracy?

4. On the experimental results. (1) This paper choses this KITTI to conduct experiments. Although the good results are shown in Table 3, the method of this paper (anonymous) is not seen on KITTI's official website. The results on the KITTI test set should be submitted to the KITTI official website, so please submit your results anonymously on the KITTI official website, otherwise the authenticity of the experimental results in this article will be seriously questioned. (2) In the 3D object detection task, more data sets have emerged, such as Waymo, nuScences, etc. This article should try on more data sets.

5. The writings should be checked carefully. There are many singular and plural problems, symbol problems, tense problems and so on. Such as need->needs in Section 2, two repeated symbols f_j below formula 1.

**Summary Of The Paper:**

 Aiming at accurately detecting objects in the 3D space from a single image, this paper proposes a new method. Experiments on KITTI dataset seems to show promising results.

**Summary Of The Review:**

This paper has many problems in experiments, especially the authenticity of the experimental results, and the completeness and richness of the experiments. And there are many writing errors in this paper.

---

> ### Author Response · Authors · 2021-11-18
> **Authors Response to Reviewer J4VF**
>
> **We thank all the valuable comments for this work. Here we mainly answer the weaknesses in the rebuttal period.**
>
>
> **Q1: about the usage of the LiDAR data in the training phase.**
>
> In the field of monocular 3D detection, applying LiDAR sweeps in the training phase is empirically allowed, and lots of works, such as [1,2,3,4,5], are designed under this setting. In this work, we adopt the setting of these works.
> Differently, these works use LiDAR sweeps as extra training labels, while this work learns the knowledge from LiDAR data after distilling by CNN.
>
>
> [1] Categorical depth distribution network for monocular 3d object detection, CVPR'21
>
> [2] Monorun: Monocular 3d object detection by reconstruction and uncertainty propagation, CVPR'21
>
> [3] Depth-conditioned dynamic message propagation for monocular 3d object  detection, CVPR'21
>
> [4] The devil is in the task: Exploiting reciprocal appearance-localization features for monocular 3d object  detection, ICCV'21
>
> [5] Accurate monocular 3d object detection via color-embedded 3d reconstruction for autonomous driving, ICCV'19
>
> **Q2: Figure 3 is very confusing.**
>
> Thanks for this suggestion, and we will update Figure 3 in the revised version.
>
> **Q3: The importance of monocular-based methods.**
>
> First, we do not think that the monocular methods are superior to LiDAR-based methods.
> In fact, both of them play important role in the perception system.
> Besides, we argue that, compared with LiDAR-based methods, the monocular methods have several irreplaceable advantages, including but not limited to:
>
> (i) camera is more cheaper and easy-to-deploy than LiDAR;
>
> (ii) the monocular models perform better in some specific scenes (e.g. LiDAR-based model is hard to detect holed object such as railings);
>
> (iii) cameras can capture richer semantic information (LiDAR focus more on spatial information) which has potential value in real-world applications (e.g. it is easier for vision-based 3D detectors to predict the fine-grained class labels, such as 'private car' or 'police car', than LiDAR-based models).
>
> Based on these reasons, the monocular methods also have promising application potential (e.g. Elon Musk announced that Tesla will develop the pure vision-based autonomous driving system) and deserve more studies.
>
> **Q4: about the experimental results.**
>
> > the method of this paper (anonymous) is not seen on KITTI's official website
>
> Under the default settings of KITTI, the submission results are invisible to the public.
> We change the corresponding setting, and the anonymous submission is visible now.
> You can check it on KITTI's official website
> ([quick link](http://www.cvlibs.net/datasets/kitti/eval_object_detail.php?&result=3f855e4cdf34ee2e9c88ab5b0a13db44d8f0fdec)).
>
> > more data sets have emerged, such as Waymo, nuScences, etc.
>
> However, it is hard for us to complete experiments on the large-scale nuScenes or Waymo in the rebuttal period due to the limitation of time/resources.
> Nevertheless, these experiments will be explored in future work.
>
> **Q5: writing errors.**
>
> Thanks for the careful review and we will fix the typos accordingly in the revised version.

---

### Official Review · Reviewer_mGRP · 2021-10-30

**Correctness:** 3
**Technical Novelty And Significance:** 2
**Empirical Novelty And Significance:** 3
**Recommendation:** 8
**Confidence:** 4

**Main Review:**

Strengths:
- The basic idea of this paper is easy to follow.
- The motivation and core contributions are clearly presented.
- The methodology is simple yet effective.
- Some experimental designs are novel and convincing. For instance, Table 5 for cross-model evaluation can validate that the localization accuracy has indeed improved a lot, and Table 6 provides a counterintuitive conclusion that the performance of the teacher model is not directly related to the performance improvement.
- The overall model achieves state-of-the-art on the KITTI benchmark while maintaining impressive efficiency.

Weaknesses:
- The main concern is that the technical novelty is limited. This paper is basically an application of knowledge distillation on monocular 3D detection while only different in terms of specific designs and conclusions due to different tasks.
- The generalization ability of this method is not discussed. For example, if the settings of LiDARs or cameras are changed, can the student network generalize to other scenarios well? If the consistency of these intrinsic and extrinsic settings is quite necessary, the application of this kind of method will also be much more limited.
- A minor problem is that the size of the KITTI dataset seems too small for now. It would be more convincing if the method can be validated on large-scale datasets, such as nuScenes and Waymo.

Minor comments:
- There are some small grammatical typos, such as “thus no extra computational cost is introduced” needs an “and” in front, and “Existing LiDAR-based models based on” needs an “are”. Please double-check, fix them and also polish the writing.
- Figure 4: The Gaussian-like mask means Gaussian weights in Eqn. (4) or only center-sampling regions with equal weights? Here it also lacks reference to FCOS [1] for 2D detection and FCOS3D [2] for monocular 3D detection.
- “Do we need depth estimation as an intermediate task”: This claim is closely related to another missing reference (it does not matter because it is really recently published), DD3D [3]. The difference in terms of method design is clear but it can be useful if an empirical comparison can be conducted.
- Table 7: It is not clearly presented the source of training data for DORN. Is it from KITTI-Depth or other datasets? Is there any data-leakage problem? Does it influence the result or conclusion in this part?
- Appendix A.3: How was the depth error computed? Is it done for top-k detection predictions or in other ways?

References:

[1] FCOS: Fully Convolutional One-Stage Object Detection. ICCV 2019

[2] FCOS3D: Fully Convolutional One-Stage Monocular 3D Object Detection. ICCVW 2021

[3] Is Pseudo-Lidar needed for Monocular 3D Object detection? ICCV 2021

**Summary Of The Paper:**

This paper proposes an approach to leveraging knowledge distillation (KD) for training image-based monocular 3D detectors. Different from the ways to incorporate the LiDAR signal in prior works, this paper takes it as the input of a teacher network and further supplements the image-based student network in terms of spatial information. Three levels of distillation, scene-level, object-level in the feature space, and object-level in the result space, are devised. Experimental results show that this method can effectively boost the performance of image-based monocular 3D detection while still maintaining outstanding efficiency.

**Summary Of The Review:**

This paper applies knowledge distillation to image-based monocular 3D detection and achieves state-of-the-art results. It can provide a potential path to better leverage LiDAR signals to boost the image-based 3D detectors while not much influencing the original design and efficiency. However, there are some concerns in terms of novelty and generalization ability. So I would vote for borderline accept temporarily.

---

> ### Author Response · Authors · 2021-11-18
> **Authors Response to Reviewer mGRP**
>
> **We thank all the valuable comments for this work. Here we mainly answer the weaknesses in the rebuttal period.**
>
> **W1: technical novelty is limited**
>
> Although knowledge distillation had been well studied in many fields,  it is rarely investigated to monocular 3D detection.
> In this work, we find knowledge distillation can effectively and efficiently introduce spatial cues to monocular models.
> More importantly, the lack of this kind of features is the main reason of
> low performance of existing methods.
> Based on this, we believe our work can provide insights to other researchers and is beneficial to the monocular 3D detection community.
>
> **W2: generalization ability of this method**
>
> Thanks for this comment.
> Here we discuss the generalization ability of the proposed method in the following aspects.
>
> First, we apply the proposed method on another baseline model (GUPNet, published on ICCV'21), and the  experimental results are shown in the following table:
>
> |    | IoU  | Easy  | Mod. | Hard |
> |  ----  | ----  | ----  | ----  | ----  |
> | GUPNet-baseline  | 0.7 | 22.76 | 16.46 | 13.72 |
> | GUPNet-ours  | 0.7 | 24.43 | 16.69 | 14.66 |
> | GUPNet-baseline  | 0.5 | 57.62 | 42.33 | 37.59 |
> | GUPNet-ours  | 0.5 | 61.72 | 44.49 | 40.07 |
>
> We can see that the proposed method also works for GUPNet, which suggests our method can generalize to other baseline models.
>
> Second, following Pseudo-LiDAR++ (ICLR'2020), we generate the simulated 32-beam/16-beam LiDAR signals and use them to train our teacher model (in the `sparse' setting).
> We show the experimental results, based on MonoDLE, in the following table:
>
> |    | Easy  | Mod. | Hard |
> |  ----  | ----  | ----  | ----  |
> | baseline | 19.29 | 15.13 | 12.78 |
> | ours-16-beam | 22.49 | 17.66 | 15.08 |
> | ours-32-beam | 23.24 | 17.71 | 15.19 |
> | ours-64-beam | 23.61 | 18.07 | 15.36 |
>
> We can see that, although the improvement is slightly reduced w.r.t. the decrease of the resolution of LiDAR signals, the proposed method significantly boosts the performances of the baseline model under all settings.
>
> Besides, note that the camera parameters of the images on the KITTI testing set are different from these of the training set,  and the good performance on the testing set suggests the proposed method can also generalize to different camera parameters.
> However, generalizing to the new scenes with different statistical characteristics is hard task for existing 3D detectors, including the image-based models and LiDAR-based models, and deserves further investigation by future works.
> We also argue that the proposed method can generalize to the new scenes better than other monocular models because ours model learns the stronger features from the teacher net.
>
> **These results and analysis had been included in the revised version (Appendix A5).**
>
> **W3: experiments on the large-scale dataset.**
>
> It is hard for us to complete experiments on the large-scale datasets in the rebuttal period due to the limitation of time/resources.
> Nevertheless, these experiments will be explored in future work.
>
> **M1: small grammatical typos**
>
> Thanks for the careful review, and we will fix these typos accordingly in the revised version.
>
> **M2: about the Gaussian weight and references**
>
> We employ equal weights for the pixels in the Gaussian region.
> In particular, we generate the Gaussian kernel using the shape of the bounding box, and the pixels whose response values surpass a predefined threshold are sampled, and then we train these samples with equal weights.
> By the way, we also tried the Gaussian weights, and the performance is almost the same as that of the scheme we adopted.
> Besides, we will add FCOS, FCOS3D as references.
>
>
> **M3: about DD3D.**
>
> Thanks for this suggestion.
> As you said, DD3D uses a totally different method to study whether the intermediate depth estimator is needed or not, and we discuss this work in the analysis of Table 7.
>
> **M4: the source of training data for DORN**
>
> We use the pre-trained DORN model, which trained from 23,488 frames on the KITTI Depth dataset.
> There is an overlap between the KITTI Depth's training set and KITTI 3D's validation set.
> Note that this data leakage problem does not affect the conclusion of Table 7.
> In particular, the proposed method (setting c in Table 7, trained from 3,712 images without data leakage) performs better than the control trial (setting b in Table 7, trained from 23,488 images with data leakage), which proves unnecessary of intermediate depth estimator.
>
>
> **M5: How was the depth error computed?**
>
> We project all the ground-truth instances into the image plane and collect the estimated depth values at corresponding positions for evaluation.

---

> > ### Comment · Reviewer_mGRP · 2021-11-29
> > **Final Rating**
> >
> > Thanks for all the insightful ideas from other reviewers and the response of the authors. From my point of view, the author provides more extensive experimental results and addresses most concerns. I would raise my rating to accept.
> >
> > I would strongly recommend the authors also include the baseline KD results in the final version (as given in the response to Reviewer ZFtD) and make a more detailed discussion about the comparison details and technical contributions thereon.

---

### Official Review · Reviewer_nFhM · 2021-10-31

**Correctness:** 3
**Technical Novelty And Significance:** 2
**Empirical Novelty And Significance:** 3
**Recommendation:** 8
**Confidence:** 4

**Main Review:**

In the following, strengths (S1-4) and weaknesses (W1-3) are detailed.

S1. Distillation pipeline. The way in which the student is trained through scene-level and object-level distillation together with the extended pseudo labels is an interesting approach to leverage the LiDAR input at training time which does not require architectural changes between student and teacher.

S2. Comparison to intermediate depth estimation tasks. Ablation experiments are conducted to question the usage of the additional task of depth estimation often used for 3D detection (also from stereo in the supplementary material). They give insight that this direction might be less effective. This might affect future network design choices.

S3. Results. The experimental evaluation of all involved components (loss functions, but also the densification) justifies their contributions. The overall accuracy of the pipeline is favourable over many state-of-the art approaches.

S4. Paper presentation. The paper is clearly motivated and easy to understand.

W1. Fair comparison. In the comparison (Tab. 3): The presented method should also get a "*" as the LiDAR signal is implicitly used for training.

W2. Minor notation information: In eqn. (3) information is missing on the index k.

W3. Some typos, grammar mistakes and minor issues make some sections of the paper hard to read. These are (in order of appearance):
"v.s."
"geometric constrain"
"then use LiDAR-based model"
"lots works"
"monocular v.s. stereo"
"results shows"
"baseline model adopt"
"and use several"
"this model achieve"
"models [are] mainly based on"
"the 2D bounding box are used"
"we can found that"
"our full model improve"
"can only takes"
point before Table 3
"Table 7 show"
"23,488 v.s. 3,712"
"depth maps v.s. noisy depth"

**Summary Of The Paper:**

The paper presents MonoDistill, a way to enhance RGB-based 3D object detection through knowledge distillation from a LiDAR-based teacher.
A 3D detection pipeline (built around MonoDLE) is trained both on RGB images and densified LiDAR input. Three main mechanism enable teacher-student transfer: On feature level, scene-level distillation aligns affinity maps between teacher and student while masked features are trained on object-level, additional masked pseudo labels are leveraged around the teacher centre predictions.
Each component is carefully ablated and experiments are conducted on the KITTI benchmark.


**Summary Of The Review:**

While the paper has minor weaknesses, I believe that they can be corrected in the course of the review process.
The strenghts (in particulat S1-3) make this work worth sharing.

---

> ### Author Response · Authors · 2021-11-18
> **Authors Response to Reviewer nFhM**
>
> **We thank all the valuable comments for this work. Here we mainly answer the weaknesses in the rebuttal period.**
>
>
> **W1: marks in Table 3**
>
> We will add the $*$ superscript of our method in Table 3 in the revised version.
>
> **W2: minor notation information**
>
> We will add corresponding text for Equation 3 in the revised version.
>
> **W3: Some typos**
>
> Thanks for the careful review.
> We will fix the typos and grammar mistakes accordingly. Besides, we will further revise corresponding text for better presentation.

---

### Official Review · Reviewer_ZFtD · 2021-11-02

**Correctness:** 4
**Technical Novelty And Significance:** 3
**Empirical Novelty And Significance:** 3
**Recommendation:** 6
**Confidence:** 5

**Main Review:**

Pros:
1.	The paper is well written. The overall pipeline is simple and the key factor of KD is well illustrated.
2.	The motivation and experiments look reasonable to me. I think this is the first paper that distills the features from the LiDAR model to the monocular model to learn depth cues, which gets good results and improvement. Experiments demonstrate the student model learns to predict better 3D location. Without inference cost, the improvement is indeed impressive despite KD being a well-known approach to improve model performance.

Cons:
1.	The main concern is about the technical improvement for distillation. As there are some papers [1] that demonstrates the effectiveness of cross-modal distillation, I think the paper should at least be compared with the baseline KD approaches and demonstrate the superiority of the proposed modules such as ``scene/object-level distillation'', whether to distill affinity map and discuss why the proposed methods work better than simple KD for monocular 3D object detection, or particularly depth estimation. Otherwise, the contribution is not enough as it cannot provide any new information for cross-modal KD. Also, the related works about KD for 2D object detection (e.g. [2]) as the main point of this paper can be used as the baseline and should be included at least.
2.	From the ablation studies, it seems that the depth cues are the key factor to affect the learning of the model. Is it possible due to the teacher network interpolates the sparse depth map and thus the network transfers better the depth supervision? I think an ablation study of dense depth map supervision can help the analysis.
3.	The performance improvement for Pedestrian and Cyclist is not so obvious. The comparison with the baseline model in KITTI validation is missed and should be supplemented.

[1] Cross Modal Distillation for Supervision Transfer. Gupta S, et al. CVPR 2016
[2] Learning Efficient Object Detection Models with Knowledge Distillation, Chen et.al. NeurIPS 2017



**Summary Of The Paper:**

This paper proposes to distill the features from a LiDAR teacher model to a monocular-based student model. To align the feature maps between the teacher and student model, the teacher model uses the same networks as the student and the only difference is that the teacher model takes the sparse/dense depth map as input. Several techniques include scene-level distillation, object-level distillation in feature/outputs, feature fusion. In the experimental part, this paper extensively ablates several key analyses including cross-modal evaluation between baseline model and full model, depth estimation.

**Summary Of The Review:**

The basic motivation is quite good and is worth studying. The current ablation studies also demonstrate the good point of cross-modal distillation. The performance seems good enough in the related works. But it’s known that cross-modal KD is helpful as shown in the Main Review (1), which is not a new enough idea. And the technical improvement of the proposed KD method over the baseline KD approach should be strengthened more. Overall, I would like to give the rating of borderline reject.

---

> ### Author Response · Authors · 2021-11-18
> **Authors Response to Reviewer ZFtD**
>
> **We thank all the valuable comments for this work. Here we mainly answer the weaknesses in the rebuttal period.**
>
> **Q1: should at least be compared with the baseline KD approaches and whether to distill affinity map and discuss why .**
>
> Thanks for this suggestion.
> In the work, we found directly enforcing the image-based model learns the feature representations of the LiDAR-based models is sub-optimal, caused by the different modalities.
> By contrast, the affinity map encodes the relative relations of the features, keeping the knowledge structure and alleviating the modality gap, which is the main reason for the effectiveness of this design.
> Besides, Table 1, 2 in the main paper report the ablations for the distillation designs under several settings, which can be regarded as baseline models.
> We will add more baselines (such as directly distilling the features in scene-level using L1/L2 loss) in the revised version.
> We will cite and discuss the relevant works.
>
> **Q2: interpolates the sparse depth map.**
>
> Thanks for this suggestion, and we design the following experiment for this ablation.
> First, we use the sparse LiDAR maps as input to train the teacher net and add an extra branch which supervised by the dense LiDAR maps.
> After that, we use this teacher model to guide the student model, keeping other settings unchanged (if there is any misunderstanding for this comment, we can further modify the experiment setting).
> The experimental results are shown in the following table:
>
> |    | Easy  | Mod. | Hard |
> |  ----  | ----  | ----  | ----  |
> | baseline  | 19.29 | 15.13 | 12.78 |
> | sparse  | 23.61 | 18.07 | 15.36 |
> | dense  | 24.31 | 18.47 | 15.76 |
> | sparse2dense  | 23.48 | 18.22 | 15.22 |
>
> We can see that adding an additional depth completion branch in the teacher can not bring significantly improve the performance of the student model.
>
> **Q3: the performance of Pedestrian and Cyclist.**
>
> The training samples of cyclists and pedestrians are significantly less than that of cars, as shown in the following table (collected on the training set).
>
>
> |    | Car  | Pedestrian  | Cyclist  |
> |  ----  | :----:  | :----:  | :----:  |
> | Instances  | 14357 | 2207 | 734 |
>
> Therefore the performances of these two categories are relatively low and hard to improve.
> Even so, the proposed method can also boost the performance of these two categories on the testing set.
> Besides, here we show the results on the validation set in the following table, which shows the proposed methods can improve the performance of the baseline model on the KITTI validation for pedestrians and cyclists.
> These results will be included in the revised version.
>
> |    |   | Easy 3D@IoU=0.25  | Mod. 3D@IoU=0.25  | Hard 3D@IoU=0.25  | Easy 3D@IoU=0.5  | Mod. 3D@IoU=0.5  | Hard 3D@IoU=0.5  |
> |  ----  | ----  | ----  | ----  | ----  | ----  | ----  | ----  |
> | Pedestrian  | Baseline | 28.91 | 23.64 | 19.69 | 6.75 | 5.23 | 4.27 |
> | Pedestrian  | Ours | 31.81 | 24.84 | 20.65 | 8.54 | 6.91 | 5.08 |
> | Cyclist  | Baseline | 20.65 | 11.72 | 10.78 | 3.44 | 1.66 | 1.60 |
> | Cyclist  | Ours | 25.03 | 12.90 | 12.41 | 5.04 | 2.38 | 2.23 |

---

> > ### Comment · Reviewer_mGRP · 2021-11-20
> > **About the revision of this paper**
> >
> > Thanks for your reply. I just wonder whether there has been some update results about the baseline KD approaches. I do not find them in the revised paper.
> >
> > I would also recommend you have a brief summary for the revision contents or mark them out such that reviewers can directly pay more attention to those places.

---

> > > ### Author Response · Authors · 2021-11-22
> > > **Authors Response to Reviewer mGRP**
> > >
> > > Thanks for the suggestion, and we had indicated the revised parts for each reviewer's main concerns. The technical details are answered in this rebuttal window, and the minor issues like typos are fixed accordiningly.

---

> > ### Comment · Reviewer_ZFtD · 2021-11-20
> > **Thanks for the authors' reply.**
> >
> > For Q1, in my view, cross-modal distillation is useful as it provides extra better supervision as proved by KD. But what I want to see is that why is this design of cross-modal distillation more profitable. For example in Table 1, the comparison of +Scene-level distillation, +Object-level distillation is obviously better than none in my view. So I think an extra comparison with some baseline KD approaches or loss choices can make the paper stronger.
> >
> > For Q2, What I thought is perhaps the improvement comes from the dense lidar supervision? Table 6 shows the different guidance for the teacher model. So I actually request the performance comparison with the model with dense depth supervision but without distillation. If the performance is similar, then distillation is not required indeed. We can just use a depth model like DORN to generate a dense depth map that provides cross-modality supervision.
> >
> > For Q3, I am not sure the improvement is quite general to other categories as results of Cyclist in Table 8 (KITTI test) get no good improvement. There are indeed performance variances due to the limited data for these two categories. It would be more convincing if the improvement can be shown stable after repeated experiments like three times.

---

> > > ### Author Response · Authors · 2021-11-22
> > > **Authors Response to Reviewer ZFtD**
> > >
> > > **Q1: compared with baseline KD methods.**
> > >
> > > Thanks for your suggestion on conducting experiments via baseline KD methods. The following table shows the experiment results of baseline KD. We directly apply L1, L2/SmoothL1 loss on feature maps to mimic spatial features. As shown in the results, The effects of baseline KD methods are limited due to the modality gap. However, modality-agnostic scene-level and object-level distillation of our design can alleviate the problem.
> > >
> > > |    | Mod. 3D@IoU=0.7  | Easy 3D@IoU=0.7  | Hard 3D@IoU=0.7  |  Mod. BEV@IoU=0.7  | Easy BEV@IoU=0.7  | Hard BEV@IoU=0.7  |
> > > |  ------  | ----  | ----  | ----  | ----  | ----  | ----  |
> > > | Baseline  | 15.13 | 19.29 | 12.78 | 20.24 | 26.47 |18.29 |
> > > | Baseline + l1loss  | 16.17 | 21.79 | 14.28 | 22.04 | 28.08 | 19.26 |
> > > | Baseline + l2loss  | 16.13 | 21.52 | 14.18 | 22.04 | 27.85 | 19.04 |
> > > | Baseline + smoothl1  | 16.12 | 21.42| 14.26 | 22.37 | 28.62 | 19.31 |
> > > | Ours  | 18.47 | 24.31 | 15.76 | 25.40 | 33.09 | 22.16 |
> > >
> > >
> > > **Q2: dense depth supervision.**
> > >
> > > Thanks for the clarification.
> > > Here we conduct the control experiment by adding a new depth estimation branch, which is supervised by the dense LiDAR.
> > > This model is trained without KD.
> > > The following table compares the performances of the baseline model, the new control experiment, and the proposed method.
> > > From these results, we can get the following conclusions:
> > >
> > > (i) additional depth supervision can introduce the spatial cues to the models, thereby improving the overall performance;
> > >
> > > (ii) the proposed method significantly performs better than the baseline model and the new control experiment, which demonstrates the effectiveness of our method.
> > >
> > > |    | Mod. 3D@IoU=0.7  | Easy 3D@IoU=0.7  | Hard 3D@IoU=0.7  | Mod. 3D@IoU=0.5  | Easy 3D@IoU=0.5  | Hard 3D@IoU=0.5  |
> > > |  ------  | ----  | ----  | ----  | ----  | ----  | ----  |
> > > | Baseline  | 15.13 | 19.29 | 12.78 | 43.54 | 57.43 | 39.22 |
> > > | Baseline + dense depth supervision  | 16.63 | 21.32 | 13.98 | 46.19 | 60.42 | 41.88 |
> > > | Ours  | 18.47 | 24.31 | 15.76 | 49.35 | 65.69 | 43.49 |
> > >
> > > **Besides, these results are included in Appendix A6 in the revised version.**
> > >
> > > **Q3: other categories**
> > >
> > > As you said, we also observed the performance variance.
> > > To get a reliable conclusion, we conduct another two rounds of experiments with different random seeds, and the results are summarized in the following table ($\pm$ captures the standard deviation over random seeds):
> > >
> > > We can see that, compared with the baseline model, the proposed method can get better performances in all experiments, which confirms the effectiveness of the proposed method for cyclists and pedestrians.
> > >
> > > |    |    |   | Easy 3D@IoU=0.25  | Mod. 3D@IoU=0.25  | Hard 3D@IoU=0.25  | Easy 3D@IoU=0.5  | Mod. 3D@IoU=0.5  | Hard 3D@IoU=0.5  |
> > > |  ----  |  ----  | ----  | ----  | ----  | ----  | ----  | ----  | ----  |
> > > | Exp1   | Pedestrian  | Baseline | 28.91 | 23.64 | 19.69 | 6.75 | 5.23 | 4.27 |
> > > | Exp2   | Pedestrian  | Baseline | 28.93 | 23.69 | 19.81 | 6.48 | 5.24 | 4.27 |
> > > | Exp3  | Pedestrian  | Baseline | 29.37 | 23.99 | 20.05 | 7.17 | 5.05 | 4.59 |
> > > |  Mean | Pedestrian  | Baseline | 29.07$\pm$0.21| 23.77$\pm$0.15 | 19.85$\pm$0.14 | 6.8$\pm$0.28 | 5.17$\pm$0.08 | 4.37$\pm$0.15 |
> > > | Exp1  | Pedestrian  | Ours | 31.81 | 24.84 | 20.65 | 8.54 | 6.91 | 5.08 |
> > > | Exp2 | Pedestrian  | Ours | 33.06 | 26.19 | 22.26 | 10.65 | 7.80 | 6.34 |
> > > | Exp3  | Pedestrian  | Ours | 31.40 | 25.55 | 20.53 | 7.65 | 5.82 | 4.55 |
> > > |  Mean | Pedestrian  | Ours | 32.09$\pm$0.71| 25.53$\pm$0.55 | 21.15$\pm$0.79 | 8.95$\pm$1.26 | 6.84$\pm$0.81 | 5.32$\pm$0.75 |
> > >
> > > |    |    |   | Easy 3D@IoU=0.25  | Mod. 3D@IoU=0.25  | Hard 3D@IoU=0.25  | Easy 3D@IoU=0.5  | Mod. 3D@IoU=0.5  | Hard 3D@IoU=0.5  |
> > > |  ----  |  ----  | ----  | ----  | ----  | ----  | ----  | ----  | ----  |
> > > | Exp1 | Cyclist  | Baseline | 20.65 | 11.72 | 10.78 | 3.44 | 1.66 | 1.60 |
> > > | Exp2 | Cyclist  | Baseline | 20.82 | 11.74 | 10.79 | 3.29 | 1.77 | 1.71 |
> > > | Exp3 | Cyclist  | Baseline | 21.72 | 12.15 | 10.74 | 4.40 | 2.12 | 1.61 |
> > > | Mean | Cyclist  | Baseline | 21.06$\pm$0.46 | 11.87$\pm$0.19 | 10.77$\pm$0.02 | 3.71$\pm$0.49 | 1.88$\pm$0.23 | 1.64$\pm$0.04 |
> > > | Exp1  | Cyclist  | Ours | 25.03 | 12.90 | 12.41 | 5.04 | 2.38 | 2.23 |
> > > | Exp2 | Cyclist  | Ours | 25.31 | 13.64 | 12.70 | 4.48 | 2.40 | 2.30 |
> > > | Exp3  | Cyclist  | Ours | 22.45 | 12.58 | 11.14 | 6.62 | 3.23 | 3.06 |
> > > | Mean | Cyclist  | Ours | 24.26$\pm$1.29 | 13.04$\pm$0.44 | 12.08$\pm$0.68 | 5.38$\pm$0.91 | 2.67$\pm$0.40 | 2.53$\pm$0.38 |
> > >
> > > **Besides, these results are included in Appendix A2 in the revised version.**

---

> > > > ### Comment · Reviewer_ZFtD · 2021-11-29
> > > > **Thanks for the supplementary experiments.**
> > > >
> > > > The supplementary experiments have resolved some of my concerns. The paper demonstrates a successful application of cross-modal KD but the technical contribution for monocular 3D detection is still limited. Discussion about the comparison and differences with some background cross-modal KD and KD for 2D detection can make the paper stronger. I tend to raise the score to 6.

---

### Official Review · Reviewer_2Mbm · 2021-11-02

**Correctness:** 3
**Technical Novelty And Significance:** 2
**Empirical Novelty And Significance:** 3
**Recommendation:** 8
**Confidence:** 3

**Main Review:**

Strengths:

(1) The end-to-end monocular 3D detection without intermediate depth estimation, which integrates depth cues relevant for 3D detection. Intermediate depth-estimation is a real bottleneck in monocular 3D object detection as shown in multiple previous work: [Ma et al 2021, Reading et al 2021, Lu et al 2021]. Hence, this contribution to infuse depth-cues directly to the monocular network via distillation is an important contribution.

(2) The method proposed to infuse depth via distillation across structured-scene space, object feature-space, and object result-space. Through ablation study it allows to deduce that the design components are important for the 3D object detection. Furthermore, during distillation, authors mainly focus on the object-features that allow the network to focus on them and this argument is also supported by the ablation study.

(3) Paper shows that the accuracy improvement compared to the baseline model mainly comes through improvement  in depth and dimension prediction of the objects. This has been shown via a valid cross-model experiment design.

Weakness:

(1) The main claim of the paper is that the depth-cues can help in 3D object detection from monocular images. However, the results are only shown for only one teacher model [Ma et al, 2019]. This puts the generality of depth-cue based distillation in question. Also, from the section “3.2 Student Model” -- the choice of Ma et al. as a baseline is not clear. Can authors please explain what would happen with the other monocular-based methods as a teacher? Can you please show a small experiment?

(2) The motivation for the scene level distillation is not clear. The first line of Sec 3.3 “First, we believe that scene information is beneficial for our task” -- However, it is not clear why? A very similar method to the proposed scene level distillation via affinity map has already been proposed and used for semantic segmentation in the past [Hou et al 2020]. Can authors please explain why this would be beneficial for the 3D detection task? Also, please cite the relevant work.

(3) “Besides, we further normalize the confidence of each predicted object using the estimated depth uncertainty (see Appendix A.1 for more details), which brings about 1 AP improvement.” -- while it is completely fine using the tricks that bring the accuracy up, this needs to be clearly stated in the results. It is not clear if the baseline model uses such normalization. If not, the comparison with the baseline model is not an egg-to-egg comparison. This needs to be clearly stated in the paper by adding accuracy of the current method without such normalization in table 3, table 4. And mentioning the accuracy improvement by infusing depth cues and confidence normalization separately Section 4.2 “Comparison with state-of-the-art methods”


Other minor points:

(1) Usually the projected LiDAR images will have empty regions (e.g. Fig. 2 have empty black regions on the top). How is this handled during training? Since, the LiDAR image in Fig.3 shows a complete image without any empty region. Also, do we need to do any pre-processing of the camera images at the inference time which requires the use of LiDAR information?

(2) Sec 4.2 Ablation Studies “3D detection performance by 3.34, 5.02, 2.98 and improve BEV performance by 5.16, 6.62, 3.87” -- here authors should compare their numbers with the confidence normalized baseline numbers, and not the raw baseline because improvements by confidence normalization are significant in this context.

(3) Sec 4.2 detailed design choice “which improves the accuracy by 0.7” → 0.7 in what? I think this number is only valid for 3D@IOU=0.7 moderate benchmark. Authors should mention that in the text.

(4) Sec 4.2 Comparison with State-of-the-art “By contrast, our method only takes 25ms to process a KITTI Image”. However, in Table 3, the runtimes show 40ms. Why this difference? I believe 40ms is the valid number since baseline (MonoDLE)  also reports 40ms runtime.

(5) Table 3. → Authors should also put “*” superscript for their own method since it requires LiDAR data while training.

Typos:

4.2 Ablation Studies: “Specifically, we can found” →  “Specifically, we found”

4.2 Comparison with state-of-the-art methods: “our method can only takes” → “our method only takes”

4.3 What the model has learned “improvement of dimension part is also considerable (e → f)” → I think authors mean (c → f) ?

4.3 Do we need depth estimation “Here we qualitatively show the information loss” → I think it is “quantitatively”, since the table-7 are just accuracy numbers?

References:
Hou, Yuenan, et al. "Inter-region affinity distillation for road marking segmentation." Proceedings of the IEEE/CVF Conference on Computer Vision and Pattern Recognition. 2020.


**Summary Of The Paper:**

In this paper, authors propose a monocular image-based 3D object detection method based on knowledge distillation. Specifically, the monocular image-based 3D object detector [Ma et al 2019] is infused with the depth cues via distilling knowledge from LiDAR-based teacher model. At the test-time, the model detects 3D objects without any intermediate depth prediction and ranks 1st on the KITTI benchmark dataset. Hence, the proposed monocular-based 3D detector is end-to-end and more accurate than the other methods in this category, without adding extra depth-estimation overhead in between.

**Summary Of The Review:**

I recommend this paper to be accepted for the ICLR conference (score: 8 good paper). Overall, the paper is well written and does extensive ablation studies for their design choices. In this paper authors proposed a monocular Image-based 3D object detection method that distills the knowledge from the LiDAR-based teacher. At the inference time  this method predicts at 25 FPS on GPU without any intermediate depth-prediction, which is critical for the real-time application such as autonomous driving.

However, there are some minor issues in some of the claims (please see above). I would like to see the authors addressing “Weaknesses” in the rebuttal.

---

> ### Author Response · Authors · 2021-11-18
> **Authors Response to Reviewer 2Mbm**
>
> **We sincerely appreciate your detailed and thoughtful reviews and hope that our response can address your concerns.**
>
> **W1: about the baseline models and generality of the method.**
>
> > the choice of Ma et al. as a baseline is not clear
>
> We choose MonoDLE as the baseline mainly based on:
>
> (i) this method is a typical 'RGB-only' method (refer to Fig. 1 (a) in the main paper);
>
> (ii) this method was published at a recently top-tier conference (CVPR'21) with source code, which supports good performance and reproducibility.
>
> > Can you please show a small experiment?
>
> Here we apply our method on another monocular detector GUPNet [1] (ICCV'21), which also meets the above requirements, and experimental results are shown in the following table:
>
> |  | IoU  | Easy  | Mod. | Hard |
> |  ----  | ----  | ----  | ----  | ----  |
> | GUPNet-baseline  | 0.7 | 22.76 | 16.46 | 13.72 |
> | GUPNet-ours  | 0.7 | 24.43 | 16.69 | 14.66 |
> | GUPNet-baseline  | 0.5 | 57.62 | 42.33 | 37.59 |
> | GUPNet-ours  | 0.5 | 61.72 | 44.49 | 40.07 |
>
> We can find that the proposed method can also boost the performances of GUPNet, which confirms the generalization of our method.
>
> **Besides, we also recommend the reviewer refer to the Appendix A5 for more experiments and analysis about generalization.**
>
> [1] Geometry Uncertainty Projection Network for Monocular 3D Object Detection. Lu et al. ICCV'21
>
> **W2: the motivation for the scene level distillation is not clear.**
>
> > The motivation for the scene level distillation is not clear
>
> The scene-level knowledge can help the monocular 3D detectors build a high-level understanding of the given images.
> More importantly, the lack of spatial cues is the major issue of monocular 3D detection.
> However, the gap in modality makes it difficult for the image-based model to learn the feature representations of the LiDAR-based model.
> In contrast, the modality-agnostic affinity map encodes the relative relations of the features, keeping the knowledge structure and alleviating the modality gap, which is the main reason for the effectiveness of this design.
>
> > A very similar method to the proposed scene level distillation via affinity map has already been proposed and used for semantic segmentation
>
>
> Besides, Hou et al. proposed an affinity map-based KD method for road segmentation, which improves the performances of the tiny models by learning the features from large models.
> We will add relevant works, including Hou's work, in our main paper.
> In this work, we mainly focus on how to effectively and efficiently introduce spatial cues to monocular models.
> Compared with the proposed KD designs, providing a view of introducing these vital features for monocular models is our main concern.
>
> **W3: about the confidence normalization.**
>
> Thanks for this comment.
> In the experiments, both the baseline model and the proposed method adopt the confidence normalization for an egg-to-egg comparison. Besides, the performance of the baseline model without confidence normalization is shown in the following table:
>
> |   | Easy  | Mod. | Hard |
> |  ----  |   ----  | ----  | ----  |
> | validation set| 17.23 | 12.26 | 10.29 |
> | testing set | 17.45 | 13.66 | 11.68 |
>
> **For a better presentation, we have emphasized this on Page 4 and added these results in Table 3,4 in the revised version to avoid misunderstanding.**
>
> **M1: about the projected LiDAR maps.**
>
> > How is this handled during training?
>
> The pixel values of the empty regions in the LiDAR maps are set to 0, and no other processing is applied in our implementation.
> Besides, we will update the illustration of LiDAR signals in the revised version to avoid misunderstanding.
>
> > do we need to do any pre-processing of the camera images at the inference time which requires the use of LiDAR information?
>
> At the inference phase, no extra pre-processing is needed.
>
> **M2:`the effects of confidence normalized**
>
> Both the baseline model and the proposed method applied confidence normalization.
> Besides, we will add the performance of the baseline model without confidence normalization to show the effect of the confidence normalization and avoid misunderstanding.
>
> **M3, M4, and M5.**
>
> Thanks for the careful review.
> As you said, the 0.7 accuracy improvement is for the 3D@IOU=0.7 moderate setting, which is the primary metric in the KITTI Benchmark, and we will mention that in the text.
> The proposed method takes 40 ms for a KITTI image, instead of 25 ms (it should be 25 FPS).
> We will add $*$ superscript for the proposed method in Table 3.
> Besides, we will fix the typos accordingly in the revised version.

---

> > ### Comment · Reviewer_2Mbm · 2021-11-29
> > **Final Rating**
> >
> > Thank You for providing extensive response. After reading through this response and others responses, I believe this paper is fit to be presented at ICLR'22. The extra experimental results are extensive and make paper a good submission. My ratings will remain at 8- good accept.

---

### Decision · Program_Chairs · 2022-01-20

**Decision:**

Accept (Poster)

**Comment:**

This paper received 5 quality reviews, with 3 of them rated 8, 1 rated 6, and 1 rated 5. In general, while there are minor concerns, the reviewers acknowledge the contribution of applying Knowledge distillation to the problem of monocular 3D object detection, and appreciate the SOTA performance on the KITTI validation and test sets. The AC concurs with these important contributions and recommends acceptance.